# INTEGRAL View of TeV Sources: A Legacy for the CTA Project

Angela Malizia [1,*,†], Mariateresa Fiocchi [2,†], Lorenzo Natalucci [2,†], Vito Sguera [1,†], John B. Stephen [1,†], Loredana Bassani [1,†], Angela Bazzano [2,†], Pietro Ubertini [2,†], Elena Pian [1] and Antony J. Bird [3]

1   INAF/OAS Bologna, Via P. Gobetti 101, 40129 Bologna, Italy; vito.sguera@inaf.it (V.S.); john.stephen@inaf.it (J.B.S.); loredana.bassani@inaf.it (L.B.); elena.pian@inaf.it (E.P.)
2   INAF/IAPS Roma, Via Fosso del Cavaliere 100, 00133 Roma, Italy; mariateresa.fiocchi@inaf.it (M.F.); lorenzo.natalucci@inaf.it (L.N.); angela.bazzano@inaf.it (A.B.); pietro.ubertini@inaf.it (P.U.)
3   School of Physics and Astronomy, University of Southampton, Southampton SO17 1BJ, UK; A.J.Bird@soton.ac.uk
*   Correspondence: angela.malizia@inaf.it
†   These authors contributed equally to this work.

**Abstract:** Investigations that were carried out over the last two decades with novel and more sensitive instrumentation have dramatically improved our knowledge of the more violent physical processes taking place in galactic and extra-galactic Black-Holes, Neutron Stars, Supernova Remnants/Pulsar Wind Nebulae, and other regions of the Universe where relativistic acceleration processes are in place. In particular, simultaneous and/or combined observations with $\gamma$-ray satellites and ground based high-energy telescopes, have clarified the scenario of the mechanisms responsible for high energy photon emission by leptonic and hadronic accelerated particles in the presence of magnetic fields. Specifically, the European Space Agency INTEGRAL soft $\gamma$-ray observatory has detected more than 1000 sources in the soft $\gamma$-ray band, providing accurate positions, light curves and time resolved spectral data for them. Space observations with Fermi-LAT and observations that were carried out from the ground with H.E.S.S., MAGIC, VERITAS, and other telescopes sensitive in the GeV-TeV domain have, at the same time, provided evidence that a substantial fraction of the cosmic sources detected are emitting in the keV to TeV band via Synchrotron-Inverse Compton processes, in particular from stellar galactic BH systems as well as from distant black holes. In this work, employing a spatial cross correlation technique, we compare the INTEGRAL/IBIS and TeV all-sky data in search of secure or likely associations. Although this analysis is based on a subset of the INTEGRAL all-sky observations (1000 orbits), we find that there is a significant correlation: 39 objects ($\sim$20% of the VHE $\gamma$-ray catalogue) show emission in both soft $\gamma$-ray and TeV wavebands. The full INTEGRAL database, now comprising almost 19 years of public data available, will represent an important legacy that will be useful for the Cherenkov Telescope Array (CTA) and other ground based large projects.

**Keywords:** keV-TeV cosmic sources; INTEGRAL legacy data base; relativistic astrophysics

## 1. Introduction

In recent years, our knowledge of the most violent phenomena in the Universe has progressed impressively thanks to the advent of new detectors for $\gamma$-ray, on both the ground and in orbit. At the furthest extremes of this observational energy window, we have now discovered more than a thousand sources in the soft $\gamma$-ray band (20–100 keV) and more than 200 in the TeV band. At low energies, operating telescopes include INTEGRAL/IBIS, Swift/BAT, and NuSTAR; the first is the one providing the most extensive and deepest view of the galactic plane where many TeV sources are located, while the other two are the most efficient in mapping and studying the extra-galactic sky.

INTEGRAL[1] [1], which is the most relevant for the present work, was designed to perform observations in the hard X-ray /soft $\gamma$-ray energy range (15 keV–10 MeV), over a $30 \times 30$ square degrees field of view (FoV) with two main instruments IBIS [2] and SPI [3] optimised for high angular and high spectral resolution, respectively. The simultaneous monitoring at soft X-ray (3–35 keV) and optical (V-band, 550 nm) wavebands is carried out with Jem-X and OMC [4,5].

The INTEGRAL Instrument Consortium comprised 26 companies that were located across Europe and NASA; it was launched on a PROTON rocket on 17 October 2002 and, after more than 19 years in orbit, all of its instruments are still fully working. The mission has been recently extended up to the end of 2022, while a further extension up to 2025 is expected. For the purpose of the present study, the most important instrument is the imager IBIS, which by combining all available observations can now reach ∼0.2–0.5 mCrab sensitivity (depending on the sky region observed) combined with good location accuracy (∼arcmin), large field of view (80 square degrees fully coded; 850 square degrees full-width at zero response), as well as good timing (∼120 μs accuracy) and spectroscopic capabilities (spectral resolution of ∼8% at 100 keV) [2].

On the other side of the spectrum, the current generation of Imaging Atmospheric Cherenkov Telescopes (IACT) include MAGIC and VERITAS, in the northern hemisphere and H.E.S.S. in the southern hemisphere. These instruments are now allowing imaging, photometry, and spectroscopy of sources of high energy radiation with good sensitivity about 10 mCrab in 50 h of observation time) combined with good angular (few arcmin) and energy ($\Delta E/E \sim$10–20%) resolution. They typically work in an energy range spanning between 50–100 GeV to about 100 TeV and they have a field of view of 3–5 degrees [6]. For comparison, CTA will be about a factor 10 more sensitive than any of these instruments; it will cover, with a single instrument, three to four orders of magnitude in energy range, from about 100 GeV to several TeV. It will also reach an angular resolutions at the arcmin level, a factor of few lower than current instruments. These characteristics, combined with high temporal resolution (on sub-minute time scales, which are currently not accessible), great pointing flexibility, and global sky coverage, will allow a major leap in the future of GeV/TeV astronomy (for more information, see Actis et al. [7], Cherenkov Telescope Array Consortium [8]).

Connecting the properties of sources that are seen at both these extremes is very important, as it allows us to discriminate between various emission scenarios and, in turn, fully understand their nature.

Combining INTEGRAL data with TeV information and MeV/GeV archival measurements should be the most appropriate approach as it could provide an unprecedented waveband coverage of more than nine orders of magnitude and could possibly allow to discriminate between the leptonic or hadronic origin of the highest energy gamma-rays. The observation of galactic objects in the TeV energy band can greatly contribute the identification of the source of cosmic rays. Based on energetic considerations, the best candidates are supernova remnants (SNRs) that are able to accelerate particles at very high energies; this will hopefully be confirmed by the observation of gamma-rays of unambiguous hadronic origin, as well as detection of neutrinos. CTA will provide a detailed view of the acceleration sites as well as information on the propagation of these high energy particles in individual sources. Therefore, the INTEGRAL archive can play an important role in the analysis of the candidate accelerators.

In the case of extra-galactic astrophysics, CTA studies of blazars can lead to firmly establishing the origin and the production mechanisms of TeV photons from relativistic jets. Additionally, in this case, the INTEGRAL database can be useful, as it will provide essential information in terms of light curves and spectra, particularly for high energy

---

1 More information on the satellite and specific instruments, as well on the the INTEGRAL Science Operations Centre (ISOC at European Space Astronomy Center ) can be found on the ESA webpage dedicated to the mission (https://www.cosmos.esa.int/web/integral/home, accessed on 3 May 2021)

peaked objects, where it will give a significant contribution in building their spectral energy distributions (SED).

Finally, CTA is expected to discover many new sources, and their identification and study can benefit from INTEGRAL current and future survey data, where unidentified and/or poorly known objects are continuously being followed-up in the X-rays and at optical/infrared wavelengths, not only for classification purposes, but also for multiwaveband characterisation.

Here, we provide an overview of the soft $\gamma$-ray counterparts of TeV sources, as observed by INTEGRAL, which is the instrument that is providing the deepest survey of the Galactic plane and centre (12.5 Msec along the Plane and up to 52 Msec in the Galactic centre at the end of the current AO, December 2021), where most TeV sources have been discovered so far. Figure 1 illustrates the potential of INTEGRAL to provide soft $\gamma$-ray information for TeV sources: it shows an image of INTEGRAL Galactic Plane Survey with some TeV source positions superimposed. A cross correlation between the TeV on-line catalogue (http://tevcat.uchicago.edu/, accessed on 3 May 2021) and our latest IBIS survey [9], indicates that around 15–20% of the very high energy (VHE) sources have a counterpart in the soft $\gamma$-ray domain; this fraction includes objects of various types, such as X-ray binaries, pulsars/pulsar wind nebulae, and blazars, as well as some still-unidentified objects. INTEGRAL images, light curves, and spectra for all these sources represent a useful source of information for current VHE observations and a strong legacy for future projects, such as the CTA.

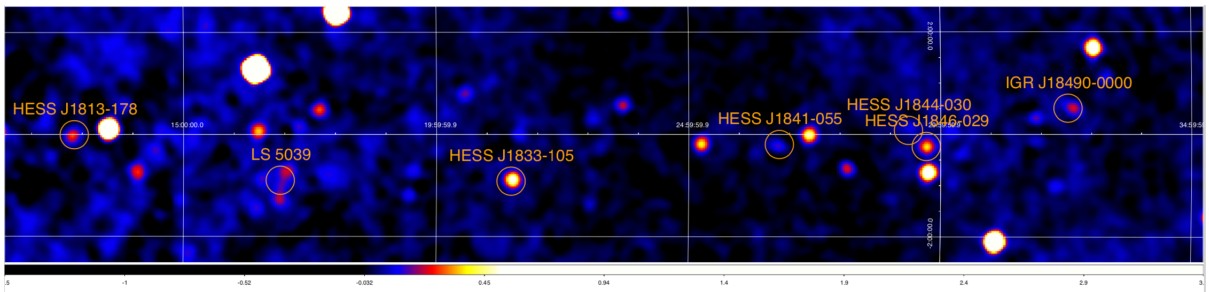

**Figure 1.** INTEGRAL image of a Galactic Plane region showing IBIS soft $\gamma$-ray sources and positions of some TeV sources.

## 2. Cross Correlation Analysis and Soft $\gamma$-ray/TeV Associations

Employing a spatial cross correlation technique that has been successfully used to identify X-ray counterparts of high energy sources (e.g., Stephen et al. [10,11]), we compare the INTEGRAL/IBIS (15 keV–10 MeV) [2] and TeV all-sky data in search of secure or likely associations.

For the INTEGRAL database, we use the 1000 orbit catalogue [9], which lists 939 sources that were detected in the 17–100 keV energy band above a 4.5$\sigma$ significance threshold; the catalogue is extracted from the mosaic of all observations that were performed by the IBIS instrument up to orbit 1000, i.e., up to the end of 2010 and, therefore, only gives a glimpse of the possible associations that are available now after 10 more years of measurements. The reason why we only concentrate on the first eight years of data is due to problems that are related to changes in the IBIS telescope performances over its lifetime (it is now almost 19 years in orbit); although these changes are small and quite expected, they nevertheless imply a correction to be made on some data analysis tools, which are, at the moment, in progress. In the future, the revised data analysis software should allow observations taken many years apart to be summed up together without compromising the correctness of the process.

For the TeV database, we used the TeV on-line catalogue considering only default and new associations, which provides a list of 229 objects (by end of 2020). We note that interesting sources, like Cygnus X-1, which is listed among candidate TeV objects, can be studied in detail at both low and high $\gamma$-ray energies by exploring the wealth of imaging, spectral, and timing data that are available in the INTEGRAL archive. Figure 2 (left side)

shows the distribution of sources in the two catalogues, where the deep coverage that is present in both along the Galactic plane is evident.

The cross correlation technique consists of simply calculating the number of TeV sources for which at least one INTEGRAL counterpart was found within a specified angular distance, out to a distance where all of the TeV sources had at least one soft $\gamma$-ray counterpart. Because the positional errors are comparable for all of the sources in either of the catalogues, the positional uncertainty was not used in the initial cross correlation, but it would come into play in the more detailed analyses after the list of possible associations was formed. To have a control group, we then created a list of *'anti-TeV sources'*, mirrored in Galactic longitude and latitude, and the same correlation algorithm applied. Figure 2 (right side) shows the results of this process. The solid curve shows that a strong correlation exists, out to about 330 arcseconds, where only 2–3 false associations are expected to be, by chance, coincidence, with the remaining being likely true ones.

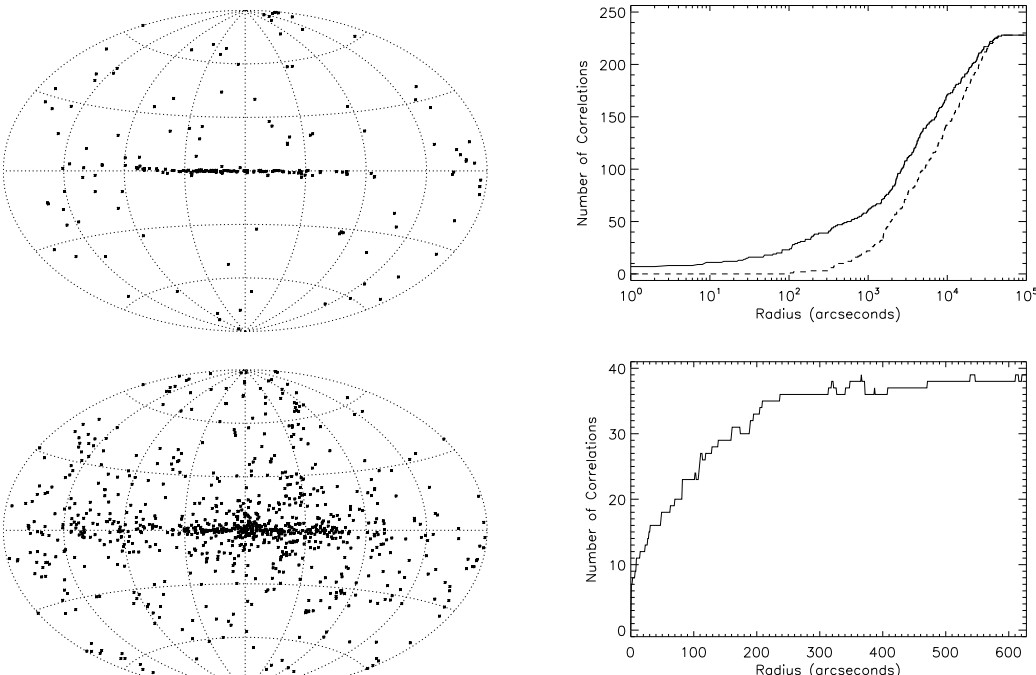

**Figure 2.** (**Left**) The distribution of TeV (**upper**) and INTEGRAL sources (**lower**) showing the galactic plane clustering. (**Upper right**) The number of TeV-INTEGRAL spatial correlations (solid curve) and the same for a 'fake' set of TeV sources with positions mirrored in latitude and longitude (dashed). (**Lower right**) The difference between the two curves showing the number of 'excess' correlations.

We found 37 possible associations, which reduces to 33 after we exclude the galactic centre/ridge regions where associations are difficult to verify and a few clearly false matches. In the top section of Table 1, we report, for each of these 33 likely associations, their names, coordinates, and classes, as reported by the TeV and INTEGRAL catalogues, respectively, being ordered by distance between the TeV and gamma excesses. It is evident from the comparison of data from both catalogues that the majority of associations are straightforward, but there are also a few cases that deserve to be studied in more detail, plus some sources (see, for example, Crab Pulsar versus its nebula) for which it is sometimes difficult to discriminate between emission regions that are seen at different wavebands. In Table 1 we also highlight those sources that are marked as extended in the TeV catalogue.

At distances greater than 330 arcsec, chance correlations become increasingly more important, although some true correlations may still be present above this threshold. A number of these extra associations are worth further analysis; they are listed as extra matches in the second part of Table 1 and are briefly discussed in the following. To enlarge the database we have also cross correlated the TeV catalogue with all sources

flagged as having been seen by ISGRI in the INTEGRAL Reference Source Catalogue (http://www.isdc.unige.ch/integral/science/catalogue, accessed on 3 May 2021). This gives us 3 extra active galactic nuclei (AGN) which have been added to the list of sources at the bottom of Table 1.

From our analysis, we find a total of 39 TeV sources having a soft $\gamma$-ray counterpart in IBIS catalogues, which suggests that around 20% of VHE objects have INTEGRAL coverage and useful data over the 20–100 keV band. This implies that the INTEGRAL legacy (to date, almost 19 years of observations as compared to the eight years of data used for the 1000 orbit catalogue) will be extremely important for any current or future TeV observations. These 39 associations cover all types of VHE objects from galactic to extra-galactic, from binaries, SNR, pulsar/pulsar wind nebulae systems to unidentified objects and AGN of various classifications. Some examples of these associations and the wealth of information that INTEGRAL can provide are described and discussed in the following sections.

**Table 1.** Association of TeV sources with INTEGRAL/IBIS.

| Name TeV | TeV Coord. [†] RA | Dec | Class TeV | Name IBIS | IBIS Coord. [†] RA | Dec | Class IBIS |
|---|---|---|---|---|---|---|---|
| Vela Pulsar | 128.836 | −45.176 | PSR | Vela Pulsar | 128.836 | −45.176 | PWN; PSR |
| Crab Pulsar | 83.633 | +22.015 | PSR | Crab | 83.633 | +22.014 | PWN; PSR |
| NGC 1275 | 49.950 | +41.512 | FR I | NGC 1275 | 49.951 | +41.512 | AGN; Sey1.5 |
| Markarian 501 | 253.468 | +39.760 | HBL | Mrk 501 | 253.468 | +39.760 | AGN; BL Lac |
| BL Lacertae | 330.680 | +42.278 | IBL | BL LAC | 330.680 | +42.278 | AGN; BL Lac |
| 1ES 1959+650 | 299.999 | +65.149 | HBL | 1ES 1959+650 | 299.999 | +65.149 | AGN; BL Lac |
| RGB J2056+496 | 314.178 | +49.669 | Blazar | IGR J20569+4940 | 314.178 | +49.669 | AGN? |
| 3C 279 | 194.046 | −5.789 | FSRQ | 3C279 | 194.047 | −5.789 | AGN; QSO |
| HESS J1846−029 | 281.600 | −2.974 | PWN | PSR J1846−0258 | 281.602 | −2.974 | SNR; PSR; PWN |
| H 1426+428 | 217.136 | +42.673 | HBL | H1426+428 | 217.136 | +42.675 | AGN; BL Lac |
| **RX J1713.7−3946** | 258.390 | −39.760 | Shell | RX J1713.7−3946 | 258.388 | −39.762 | SNR |
| **Crab** | 83.629 | +22.012 | PWN | Crab | 83.633 | +22.014 | PWN; PSR |
| HESS J1943+213 | 295.979 | +21.302 | HBL | IGR J19443+2117 | 295.984 | +21.307 | AGN; BL Lac? |
| **HESS J1813−178** | 273.400 | −17.840 | PWN | IGR J18135−1751 | 273.397 | −17.833 | SNR; PSR; PWN |
| **Cen A** | 201.360 | −43.012 | FR I | CEN A | 201.365 | −43.019 | AGN; Sey2 |
| PKS 1510−089 | 228.217 | −9.106 | FSRQ | QSO B1510−089 | 228.211 | −9.100 | AGN; QSO |
| 4C +21.35 | 186.227 | +21.379 | FSRQ | QSOB1222+216 | 186.226 | +21.366 | AGN; QSO |
| HESS J1833−105 | 278.396 | −10.572 | PWN | SNR 021.5−00.9 | 278.396 | −10.558 | SNR; PSR; PWN |
| **HESS J1808−204** | 272.155 | −20.427 | UNID | SGR 1806−20 | 272.164 | −20.411 | SGR; T |
| PSR B1259−63 | 195.705 | −63.831 | Binary | PSR B1259−63 | 195.750 | −63.833 | PSR; Be |
| LS 5039 | 276.563 | −14.820 | Binary | LS5039 | 276.563 | −14.848 | HMXB; NS |
| LS I +61 303 | 40.142 | +61.257 | Binary | GT 0236+610 | 40.132 | +61.229 | HMXB |
| Tycho | 6.340 | +64.130 | Shell | 4U 0022+63 | 6.321 | +64.159 | SNR |
| **IGR J18490−0000** | 282.233 | −0.041 | PWN | IGRJ18490−0000 | 282.257 | −0.022 | PWN; PSR |
| Cas-A | 350.808 | +58.807 | Shell | Cas A | 350.866 | +58.812 | SNR |
| Mrk 421 | 166.079 | +38.195 | HBL | Mrk 421 | 166.114 | +38.209 | AGN; BL Lac |
| **MSH 15−52** | 228.529 | −59.158 | PWN | PSR B1509−58 | 228.477 | −59.138 | PSR; PWN |
| SNR G054.1+00.3 | 292.633 | +18.870 | PWN | PSR J1930+1852 | 292.585 | +18.896 | PSR |
| HESS J1844−030 | 281.172 | −3.093 | UNID | AX J1844.7−0305 | 281.158 | −3.144 | ? [(*)] |
| **HESS J1841−055** | 280.229 | −5.550 | UNID | J18410−0535 | 280.280 | −5.570 | HMXB; XP |
| 1ES 1218+304 | 185.360 | +30.191 | HBL | SWIFT J1221.3+3012 | 185.343 | +30.136 | AGN; BL Lac |
| 1ES 0033+595 | 8.820 | +59.790 | HBL | 1ES 0033+595 | 8.969 | +59.835 | AGN; BL Lac |
| Eta Carinae | 161.146 | −59.666 | Binary | Eta Carinae | 161.196 | −59.755 | XB |
| **HESS J1837−069** | 279.410 | −6.950 | PWN | AX J1838.0−0655 | 279.508 | −6.904 | PSR; PWN |
| **Kookaburra(PWN)** | 215.038 | −60.760 | PWN | IGR J14193−6048 | 214.821 | −60.801 | PSR; PWN |
| **HESS J1616−508** | 244.100 | −50.900 | PWN | PSR J1617−5055 | 244.372 | −50.920 | PSR |
| S5 0716+714 | 110.472 | +71.343 | IBL | QSO B0716+714 | 110.366 | +71.333 | Blazar; 1 |
| TXS 0210+515 | 33.575 | +51.748 | HBL | Swift J0213.7+5147 | 33.564 | +51.704 | ? [(*)] |
| RX J1136.5+6737 | 174.125 | +67.618 | HBL | SWIFT J1136.7+6738 | 174.241 | +67.627 | Blazar; 1 |

[†] Coordinates are at J2000; positional uncertainties for the TeV coordinates can be found at http://tevcat.uchicago.edu/, accessed on 3 May 2021, while, for those of IBIS coordinated, are listed in Bird et al. [9], Mereminskiy et al. [12] ; [(*)] unclassified source.

## 3. INTEGRAL/TeV Associations

### 3.1. Supernovae, Pulsars, and Pulsar Wind Nebulae

Supernova Remnants (SNR) and Pulsar Wind Nebulae (PWN) are historically known to be sites of Cosmic Ray (CR) acceleration [13]. In addition, the detection of synchrotron emission from keV to TeV has provided further evidence that charged leptons or hadrons can be accelerated up to TeV energies, possibly via diffusive shock acceleration processes. In the last two decades, data collected with instruments from space (e.g., AGILE, FERMI, INTEGRAL) and ground (e.g., H.E.S.S., MAGIC, VERITAS), providing imaging and spectral capability, have permitted new insights into the acceleration mechanisms responsible for the production of the high energy photons to be obtained. In spite of the good sky coverage and spectral resolution, leptonic vs. hadronic models are still under debate. One more complication in the understanding of the high energy production mechanisms of the sources detected in the GeV-TeV range, is the presence (or not) of a pulsar that is generated at the time of the SN collapse, which is, in general, off-set, due to proper motion, with respect to the SNR centre. The pulsar, with its strong magnetic field and spin, often generates additional component(s) in some of the SNR spectral emission.

### 3.1.1. SNR View at Soft $\gamma$-ray and TeV Energies

It is well known that the expanding shells of supernova remnants are the sites of cosmic rays accelerated up to PeV energies. This process manifests itself in intense non-thermal radiation spanning many decades in photon energies from the radio to the TeV wavebands. INTEGRAL/IBIS offers the opportunity to cover the relatively unexplored soft $\gamma$-ray window in at least some bright objects. Three SNR-shell associations are found by the cross correlation analysis: Tycho, Cas-A, and RX J1713.7$-$3946, with the last showing extended emission in the form of a ring like configuration. The study of the shape of the broad band spectrum as well as the morphology of these three objects measured by INTEGRAL in conjunction with other observatories, provide clues for our understanding of the details of cosmic ray acceleration and the radiation mechanisms at work in the expanding shell of these remnants.

Figure 3 compares the INTEGRAL/ISGRI spectra of the Tycho and Cas-A SNR in the 20–150 keV band summing the data related to all observations performed during the first 1500 satellite orbits. For Tycho (black points in Figure 3) a simple power law with photon index $\Gamma = 2.2 \pm 0.5$ and a 20–150 keV flux of $1.2 \times 10^{-11}$ erg cm$^{-2}$ s$^{-1}$, fits reasonably well the high energy spectrum (errors in Figure 3 and elsewhere include statistical and systematic uncertainties). However, in the case of Cas-A (red points in Figure 3), a steeper power law of $\Gamma = 3 \pm 0.1$ (20–150 keV flux of $4.9 \times 10^{-11}$ erg cm$^{-2}$ s$^{-1}$) is insufficient to fit the data mainly due to the presence of a significant excess around 70–90 keV, which we attribute to the presence in the source of Titanium-44 (44Ti) decaying lines, likely located at 68 and 78 keV. Indeed, the addition of a single Gaussian line is highly required by the data at a significance level greater than 99.99%, but with the data used in the present work we are not able to fully characterise the 44Ti line complex, due to the poor energy resolution of the average IBIS spectra.

A more detailed analysis of the Tycho spectrum over a broader (3–100 keV) energy band was presented by Wang and Li [14], who found a two component model fit comprising thermal bremsstrahlung with kT~0.8 keV plus a power-law with $\Gamma \sim 3$, i.e., not much different to our fit when considering the limited energy band considered here. Based on the diffusive shock acceleration theory, this non-thermal emission, together with radio measurements, implies that in this remnant protons are accelerated up to hundreds of TeV.

The broad-band spectrum for Cas-A (thermal bremsstrahlung with kT~0.8 keV plus a power-law with $\Gamma \sim 3.2$) is similar to our previous result for the high energy spectrum; in this case, Renaud et al. [15], using INTEGRAL data, were also able to resolve the two 44Ti decaying lines at 68 and 78 keV and measure a 44Ti yield from the SNR explosion of $10^{-4}$ solar masses. Furthermore, the continuum emission seems to extend beyond 100 keV. Cas-A has been observed with the MAGIC telescope [16], its spectrum in con-

junction with the Fermi-LAT one, shows a clear turn-off ($4.6\sigma$) at the highest energies in the 60 MeV to 10 TeV energy band, which can be described with an exponential cut-off at $E_c = 3.5(^{+1.6}_{-1.0})_{stat}(^{+0.8}_{-0.9})_{sys}$ TeV. This is expected in the diffusive shock acceleration theory, which predicts a slowly decreasing spectral shape for the synchrotron radiation. This result implies that, even if all the TeV emission was of hadronic origin, Cas-A could not be a PeVatron at its present age [16].

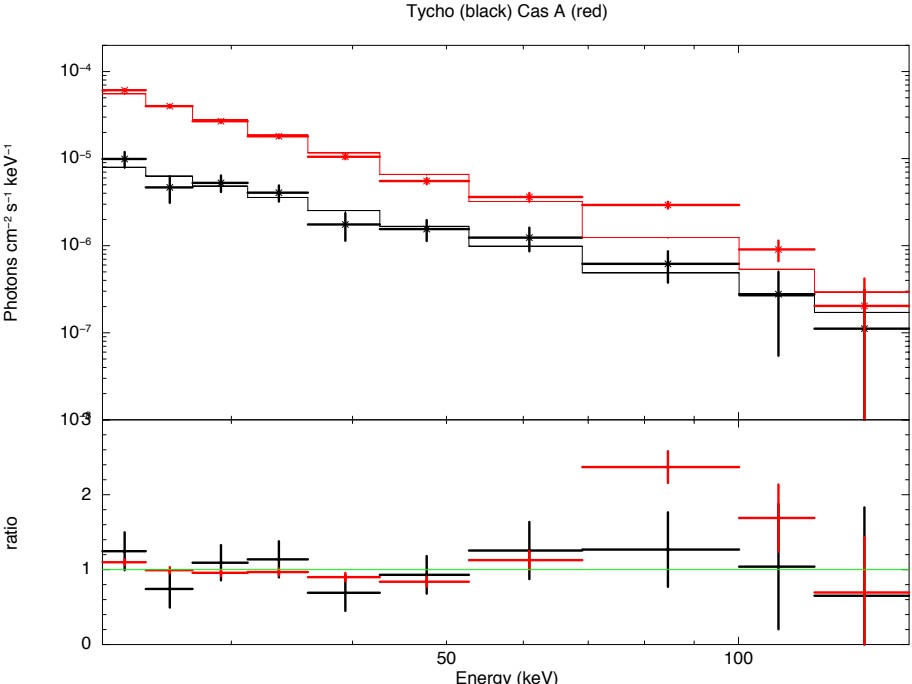

**Figure 3.** INTEGRAL/IBIS unfolded spectra and the data to model ratio of the Tycho (black) and Cas-A (red) over the 20–150 keV energy range. A simple power law fits well the data of Tycho but it is not sufficient for Cas A data where an excess around 70–90 keV is clearly detected (see text).

The supernova remnant RX J1713.7$-$3946 provides, on the other hand, a nice example of the imaging capability of INTEGRAL in the case of extended objects, since IBIS has, for the first time, resolved its spatial structure in soft $\gamma$-rays [17]. Figure 4a,b shows the colour-coded image of RX J1713.7$-$3946 obtained in the 17–60 keV energy band: a clear ring-like structure with $\sim$24 arcmin radius is evident. Superimposed in green are the surface brightness contours in the soft (0.1–2.4 keV) X-ray band mapped with ROSAT (top panel, Pfeffermann and Aschenbach [18]) and in TeV $\gamma$-rays that were obtained with the HESS telescope (bottom panel Aharonian et al. [19]). The similarity of the images in soft X-rays, soft $\gamma$-rays and at TeV energies is striking, especially when considering the nine orders of magnitude coverage in photon energies. The X-ray emission of RX J1713.7$-$3946 is most likely dominated by synchrotron radiation of electrons in the shell regions [20] accelerated up to multi-TeV energies at the supernova shock [21].

More recently, Kuznetsova et al. [22] presented a more detailed analysis of the shell morphology, as seen by IBIS, further highlighting the presence of two extended hard X-ray sources that are spatially consistent with the northwest and southwest rims of RX J1713.7–3946, i.e., the brightest parts of the SNR in Figure 4. Interestingly, Sano et al. [23]) found a good correlation between the X-ray intensity map and the presence of major CO/HI clumps with mass greater than 50 solar masses interacting with the shock waves in the SNR. The magneto-hydrodynamic numerical simulations show that the interaction between the shock waves and the clumps generates turbulence that enhances the magnetic field and synchrotron X-rays at the shocked surface of the clumps, and the enhanced turbulence and/or magnetic field re-accelerates electrons to higher energies, providing TeV emission. Thus, RX J1713.7$-$3946 represents a unique laboratory for the study of a

core-collapse SNR that emits bright TeV $\gamma$-rays and synchrotron X/soft $\gamma$-rays that are caused by cosmic rays, in addition to interactions with interstellar gas clouds.

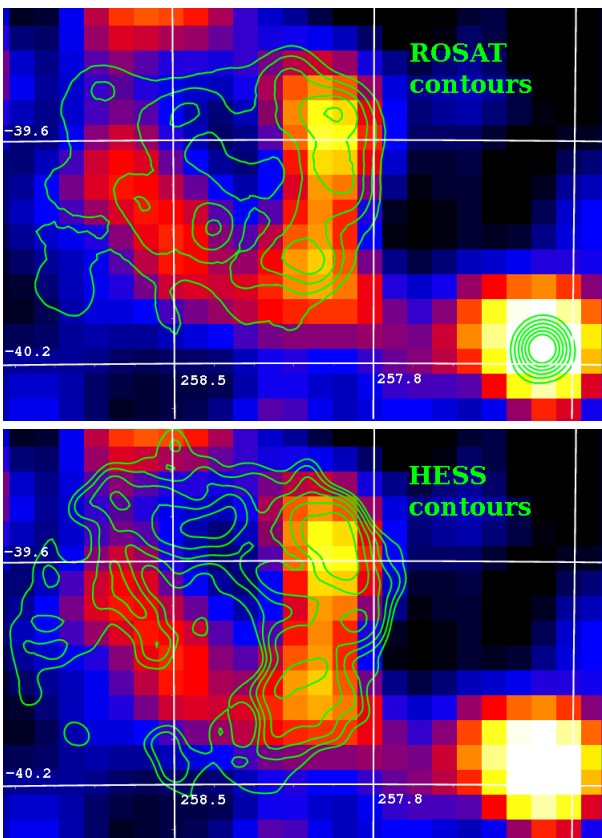

**Figure 4.** Colour-coded image of RX J1713.7−3946 in the 17–60 keV energy band obtained with INTEGRAL/IBIS with superimposed in green the surface brightness contours in the soft (0.1–2.4 keV) X-ray band mapped with ROSAT (top panel) and in TeV gamma-rays obtained with the H.E.S.S. telescope (bottom panel). The coordinates are in RA/Dec. INTEGRAL picture of the month for April 2008 (https://www.cosmos.esa.int/web/integral/pom-archive, accessed on 3 May 2021) and Krivonos et al. [17]).

### 3.1.2. Soft $\gamma$-ray Pulsars/PWN with TeV Emission

Among the several pulsar/PWN systems emitting at high energies, only four pulsars are reported in the TeV catalogue, and two of them are also clearly detected by INTEGRAL: Vela and Crab. They have both been extensively studied at soft $\gamma$-ray energies and their broad band properties, including the VHE emission, is discussed in a review conducted by Kuiper and Hermsen [24]. Regarding instead PWN, approximately 11 such systems are detected in common by INTEGRAL and TeV telescopes (see Table 1), the uncertainty being due to a few sources where the IBIS/TeV association is not straightforward. For example, the PWN that is associated with the Vela pulsar (named Vela X) has not been picked up by the cross correlation analysis, probably due to the complex morphology of this system at very high energies.

Indeed, an extended emission of about 0.8 degrees radius, located south of the Vela pulsar, has been observed with H.E.S.S. [25]. From observations with INTEGRAL/IBIS, Mattana et al. [26] claimed the detection of a spatially extended emission above 18 keV, after the subtraction of the main radiation from the pulsar. The morphology of this emission appears less extended than the source observed at TeV energies and it is consistent with the size of the X-ray cocoon observed at lower energies (see e.g., Mangano et al. [27], Slane et al. [28]. This suggests that INTEGRAL has actually observed both the pulsar and emission associated with the Vela-X PWN.

Other examples of complex systems are those of IGR J14193−6048/Kookaburra and HESS J1616−508/PSR J1617−5055. In the first case two close-by, distinct TeV sources have been observed by H.E.S.S. in the Kookaburra complex (HESS J1420−607 and HESS J1418−609, [29], see left panel of Figure 5). INTEGRAL detects a source, named IGR J14193−6048, in between these two TeV objects, although its position is shifted towards HESS J1420−607 (Figure 5, right panel). The analysis of the low energy X-ray data from Chandra suggests that the most likely identification for the INTEGRAL source is PSR J1420−6048, the pulsar that powers HESS J1420−607, although its location is barely inside the 99% IBIS error circle [30]. A soft $\gamma$-ray source that is associated with PSR J1420−6048 is also reported in the BAT 105 month catalogue [31], providing further evidence that this system (pulsar plus its PWN) is detected in the INTEGRAL energy range. The spatial coincidence observed between HESS J1420−607 and IGR J14193−604 (right part of Figure 5) further indicates that INTEGRAL also detected this TeV object.

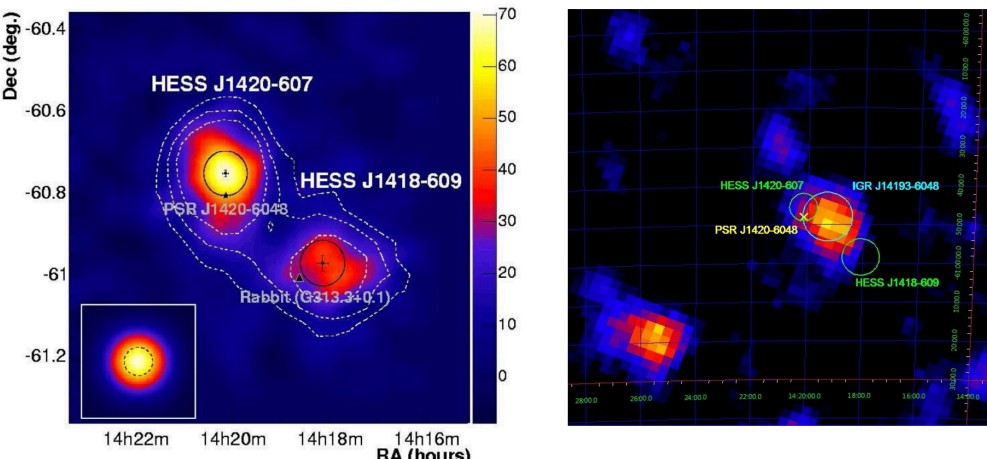

**Figure 5.** (**Left panel**): HESS image of the Kookaburra complex showing two distinct sources coincident with a PWN surrounding the pulsar PSR J1420−607 and the "Rabbit" nebula (from Aharonian et al. [29]). (**Right panel**): IBIS image of the complex in soft gamma-rays (20–100 keV). The extent of the two HESS sources (green circles) are shown together with the position of PSR J1420−607 (yellow X) and the IBIS 99% error circle for IGR J14193−6048.

A mirror situation occurs in the case of HESS J1616−508/PSR J1617−5055: the pulsar and its PWN are certainly the counterpart of the INTEGRAL source, but not necessarily that of the object emitting at TeV energies. Despite being identified with a PWN in the TeV catalogue, HESS J1616−508 is not firmly associated with any known counterpart at other wavelengths and can, therefore, be considered as still unidentified. PSR J1617−5055 is one of the possible counterparts, and its association with the TeV object has been extensively discussed by Landi et al. [32]; it is also the only pulsar in the region that is energetic enough to power a relic PWN, despite it being offset from the centre of the TeV source by ∼9 arcminutes. However, since there is no strong and convincing evidence linking other sources in the field (SNR, star cluster, etc.) to HESS J1616−508 [33,34], we consider this association to be valid for further consideration.

In summary, we find at least 11 PWN systems in common between the INTEGRAL survey catalogue and the TeV surveys; interestingly, half of them are located in the Scutum arm region of our Galaxy. In Table 2, we collect for each of these 11 associations some relevant parameters both at soft and very high gamma-ray energies, including the system distance, PWN offset from the pulsar, if emission for either component (Pulsar/PWN) is present, the photon index and luminosity both in the 20–100 keV and 0.1–10 TeV energy bands. The systems are relatively nearby (within 10 kpc), with Vela being the closest.

Systems with Pulsar/PWN offsets are also observed, namely AX J1838.0−0655 and PSR J1617−5055. From Table 2, it is clear that, while at soft $\gamma$-ray energies, the emission

is most likely due to the contribution of both the pulsar and its nebula, at TeV energies the nebula dominates in most objects, except for the case of Crab and Vela, where pulsed emission is clearly observed and independently reported. However, we note that, also in the 20–100 keV waveband, the PWN is a significant component of the 20–100 keV emission in approximately half of the systems; their contributions, typically in the range 20–30% of the total soft $\gamma$-ray luminosity, can reach higher values up to 50% in some sources [35].

A comparison between the photon indices obtained from INTEGRAL/IBIS and TeV observations indicates that PWN spectra steepen going from the soft $\gamma$-ray to the TeV waveband. The emission spectra of PWN can be broadly characterised by a double-peak structure: a low-energy peak that is produced by synchrotron radiation of electrons and a high energy hump due to Inverse-Compton (IC) up-scattering of soft-photon fields by plasma particles. As is evident in some PWN spectral energy distributions [36], the X-ray/soft $\gamma$-ray spectra mark the final part of the first peak, while the TeV emission defines the end part of the second peak; therefore, combining INTEGRAL and TeV data allows for us to properly model the source SED and estimate fundamental physical parameters. Finally, we note that, for our PWN systems, the ratio between TeV and soft $\gamma$-ray luminosity (under the assumption that the latter is 70% due to the pulsar and 30% due to the PWN) lies typically in the range 0.1–2.

**Table 2.** IBIS/TeV PWN parameters.

| Source | Dist (kpc) | Offset (pc) | IBIS (ref) PSR/PWN | TeV (ref) PSR/PWN | IBIS $\Gamma$ | $L_\gamma$ | TeV $\Gamma$ | $L_{TeV}$ |
|---|---|---|---|---|---|---|---|---|
| PSR J1846−0258 | 5.8 | <2 | (y)/y (1) | -/y (6) | $1.72 \pm 0.11$ | 160 | $2.41 \pm 0.09$ | 6.0 |
| IGR J18135−1751 | 4.7 | <2 | (y)/y (2) | -/y (6) | $1.84 \pm 0.13$ | 85 | $2.07 \pm 0.05$ | 19.0 |
| SNR 021.5−00.9 | 4.1 | <2 | (y)/y (3) | -/y (6) | $2.00 \pm 0.10$ | 107 | $2.42 \pm 0.10$ | 2.6 |
| PSR B1509−58 | 4.4 | <4 | y/(y) (4) | -/y (6) | $1.78 \pm 0.03$ | 370 | $2.26 \pm 0.03$ | 52.1 |
| PSR J1930+1852 | 7.0 | <10 | y/(y) (2) | -/y (6) | $2.0 \pm 0.5$ | 75 | $2.60 \pm 0.30$ | 5.5 |
| IGRJ18490−0000 | 7.0 | <10 | y/(y) (2) | -/y (6) | $1.57 \pm 0.16$ | 140 | $1.97 \pm 0.09$ | 12.0 |
| AX J1838.0−0655 | 6.6 | 17 | y/(y) (2) | -/y (6) | $1.69 \pm 0.12$ | 180 | $2.54 \pm 0.04$ | 204.0 |
| IGR J14193−6048 | 5.6 | 5.1 | ?/? | -/y (6) | | ∼10–20 | $2.20 \pm 0.05$ | 44.0 |
| PSR J1617−5055 | 6.5 | 17.6 | y/(y) (2) | -/y?(7) | $1.4 \pm 0.2$ | 102 | $2.34 \pm 0.06$ | 86.7 |
| Crab | 2.0 | ≤1 | (y)/y (5) | y/y (8) | $2.08 \pm 0.02$ | 7700 | $2.10 \pm 0.04$ | 86.5 |
| Vela Pulsar | 0.3 | 2.4 | (y)/y (5) | y/y (8) | $1.98 \pm 0.04$ | 1.24 | $1.89 \pm 0.03$ | 0.18 |

"y" indicates if a component (PSR or PWN) is present; y in parenthesis refers to the less prominent component. $L_\gamma$, $L_{TeV}$ are the 2–100 keV and 0.1–10 TeV luminosities in units of $10^{33}$ erg/sec. References: (1) McBride et al. [37]; (2) Kuiper and Hermsen [24]; (3) de Rosa et al. [35]; (4) Forot et al. [38]; (5) Mattana et al. [39]; (6) HESS Collaboration et al. [40]; (7) Hare et al. [33]; (8) TeV catalogue.

In Table 3, we collect data that are related to the pulsars that power the wind nebulae seen by both INTEGRAL and TeV telescopes: in particular, we list their age, pulsar period, spin down luminosity, and photon index, as measured in the X-ray band. From this Table, it is evident that the characteristic age of this specific pulsar sample, ranges from 0.7 kyr (PSR J1846−0258) up to 42.8 kyr (PSR J18490−0000), thus representing a very young population, well below 50 ky. They are all fast rotators with spin-periods between 33.5 and 324 ms, most of them (6) even spinning well below 100 ms. Moreover, the population is very energetic with spin-down powers above $5 \times 10^{33}$ erg s$^{-1}$ (the lowest value is measured for the pulsar that is associated with AX J1838.0−0655 system).

Assuming again that the soft $\gamma$-ray emission is only 70% due to the pulsar, the remaining part being related to the PWN, we obtain that the pulsar soft $\gamma$-ray luminosity is typically in the range 0.1–1% of the spin down luminosity. We also note that the photon index measured from the pulsar in X-rays is typically harder that seen by INTEGRAL ($\Delta \Gamma \sim 0.5$), a further indication of the contribution of the PWN to the total soft gamma-ray emission, since the observed steepening could be explained as synchrotron cooling away from the pulsar and in the outskirts of the PWN.

Finally, future long (>Ms) observations of these objects with INTEGRAL, thanks to its spectral, imaging, and timing capabilities, will be very useful in order to disentangle the

contribution from the different parts of these systems, allowing for the detection of various spectral components, and enabling breaks to be identified: see, for example, the case of MSH 15–52 [38].

**Table 3.** Characteristics of the Pulsars associated with IBIS/TeV PWN.

| System Name | Age (ky) | Period (ms) | Edot (erg/s) | X-ray Index |
|---|---|---|---|---|
| PSR J1846−0258 | 0.73 (1) | 324 (3) | 36.91 (1) | 1.2 (3) |
| IGRJ18135−1751 | 5.6 (1) | 44.7 (3) | 37.75 (1) | 1.3 (3) |
| SNR 021.5−00.9 | 4.85 (1) | 61.8 (4) | 37.53 (1) | 1.47 (5) |
| PSR B1509−58 | 1.56 (1) | 150 (3) | 37.23 (1) | curved (3) |
| PSR J1930+1852 | 2.89 (1) | 136 (3) | 37.08 (1) | 1.21 (3) |
| IGRJ18490−0000 | 42.90 (1) | 38.5 (3) | 36.99 (1) | 1.37 (3) |
| AX J1838.0−0655 | 22.7 (1) | 70.5 (3) | 36.74 (1) | 1.1 (3) |
| IGR J14193−6048 | 13.00 (1) | 68 (3) | 37.00 (1) | 0.5 (3) |
| PSR J1617−5055 | 8.15 (2) | 69 (3) | 37.20 (1) | 1.42 (3) |
| Crab | 1.24 (2) | 33.5 (3) | 38.67 (2) | curved (3) |
| Vela Pulsar | 11.3 (1) | 89 (3) | 36.84 (1) | 1.1 (3) |

References: (1) HESS Collaboration et al. [40] but see ATNF Pulsar Catalogue (http://www.atnf.csiro.au/people/pulsar/psrcat/, accessed on 3 May 2021) references for detailed information; (2) Mattana et al. [39]; (3) Kuiper and Hermsen [24]; (4) Camilo et al. [41]; (5) Matheson and Safi-Harb [42].

### 3.2. Binaries from keV to TeV

Gamma-ray binaries are systems that emit the dominant part of their non-thermal emission in the $\gamma$-ray domain and at VHE. They are still a rare class of objects that consist of a stellar mass compact object (either a non-accreting neutron star or an accreting black hole) characterised by their broad-band emission ranging from radio to VHE. Contrary to the X/soft $\gamma$-ray regimes, where about 300 sources have been listed [9], only a very few cases, about 11 (online TeV catalogue), have been firmly identified at VHE, and not all of them have been clearly identified in the soft $\gamma$-ray band. Indeed from our cross correlation only four sources satisfy our association criteria, namely PSR B1259−63, LS 5039, Eta Carinae, and LS I + 61 03.

The mechanisms for the production of TeV photons can have different physical origins. In the microquasar scenario, non-thermal particle acceleration processes occur in the jet of an accreting compact object [43,44], while, in the pulsar binary scenario, the particle acceleration is the result of the shock between the stellar and the pulsar winds [45]. In both models, the observed non-thermal emission can be derived from a hadronic or leptonic primary population. In a leptonic scenario, the TeV emission is the result of inverse-Compton scattering of electrons accelerated in the jet or in the shock region, for microquasars or pulsar binaries, respectively.

Multiple observations have been performed from radio to soft $\gamma$-rays up to VHE for this small class of $\gamma$-ray binaries, but the lack of simultaneous long term monitoring over a very broad band, including X-rays and $\gamma$-rays, prevents us from understanding the nature of the compact object (firmly known only for PRS B1259−63) and the physical processes that are responsible for the particle acceleration. In the following, we briefly review each object found by the cross correlation analysis. We analyse the average INTEGRAL/IBIS spectra of the three most statistically significant detected binaries by summing the data of all the observations performed during the first 1500 INTEGRAL orbits. Figure 6 shows the spectra and residuals with respect to a simple power-law model, in which we compare the IBIS spectra of the three binaries in the 20–200 keV energy range: LS 5039 in red, PRS B1259−63 in black, and LS I +61 303 in green. For LS 5039, a broken power-law is needed to fit high energy data well. Fit results are reported in Table 4, showing that the photon index spans from ∼0.8 for LS 5039 to 1.7 for PSR B1259−63. The steeper power law corresponds to the lower flux ($\sim 2.1 \times 10^{-11}$ erg cm$^{-2}$s$^{-1}$) and the harder power law to the higher flux ($\sim 7.8 \times 10^{-11}$ erg cm$^{-2}$s$^{-1}$), in agreement with class behaviour.

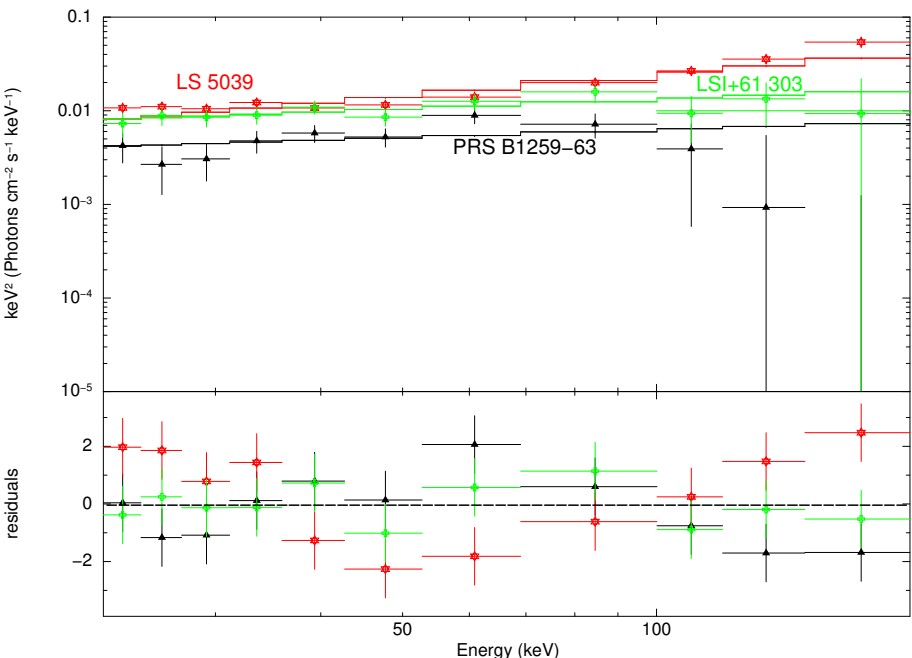

**Figure 6.** The top panel shows the IBIS/INTEGRAL spectra and models for the fit with a simple power law model of PSR B 1259−63 (red), LS5039 (green), and LSI +61 303 (black). The bottom panel shows the residuals in terms of sigmas; colours are the same as in the top panel.

**Table 4.** Results of the INTEGRAL/IBIS and SWIFT/XRT average spectra. Fluxes are in 20–200 keV and 2–10 keV energy band for INTEGRAL/IBIS and SWIFT/XRT, respectively. The SWIFT/XRT spectral fit was performed including a galactic column density.

| Name | Instrument | Model [1] | Γ | $E_{break}$ keV | Γ | Flux $10^{-11}$ erg cm$^{-2}$ s$^{-1}$ | $\chi^2_{red}$/d.o.f. |
|---|---|---|---|---|---|---|---|
| LS 5039 | IBIS | BKN | $1.9 \pm 0.3$ | $53 \pm 13$ | $0.8 \pm 0.3$ | $7.8 \pm 0.6$ | 0.6/7 |
| LS5039 | XRT | PL | $1.5 \pm 0.4$ | . . . | . . . | $0.7 \pm 0.3$ | 1.1/4 |
| PSR B1259−63 | IBIS | PL | $1.7 \pm 0.3$ | . . . | . . . | $2.1 \pm 0.4$ | 1.3/9 |
| LS I +61303 | IBIS | PL | $1.6 \pm 0.3$ | . . . | . . . | $4.3 \pm 0.5$ | 0.6/9 |

[1] BKN means a broken power-law and PL a simple power law model.

### 3.2.1. LS 5039

LS 5039 is a variable and periodic γ-ray source [46–51], having the shortest orbital period among the γ-ray binaries (3.9 days). At first, it was considered to be a microquasar [51], then the source showed evidence of X-ray pulsations with a period of 9s suggesting that LS 5039 hosts a pulsar, although no radio pulsations have been reported to date [52,53].

Each of the models proposed to reproduce the observed SED of LS 5039, involving a microquasar [54] or a non-accreting pulsar [55], fail in describing some of its features. The recent model that was proposed by Molina and Bosch-Ramon [56] reproduces the observed X-ray and VHE γ-ray emission well, but it fails to properly account for the γ-ray modulation. The non-thermal emission mechanism is still unknown, multiple models [57–60]) have been proposed to fit the LS 5039 SED, but all fail to reproduce some of the spectral features, so it would appear that there is a need for more complex models to describe the observed emission of this source. The INTEGRAL/IBIS observations show an orbital variability in the soft γ-ray band (25–200 keV) in phase with the modulation detected at TeV energies by H.E.S.S. [61], suggesting that the emission could originate from the same region, possibly from particles produced in the same acceleration process. During the phase when the source is detected, it has a flux of $\sim 3.5 \times 10^{-11}$ erg cm$^{-2}$s$^{-1}$ and the spectrum (exposure of 3 Ms) is well fitted with a power law with Γ ∼ 2.0.

The INTEGRAL/IBIS spectrum of LS 5039 is fitted with a power law with $\Gamma \sim 1.3$ and $F_{20--200keV} \sim 7 \times 10^{-11}$ erg cm$^{-2}$s$^{-1}$. Residuals that are plotted in Figure 6 show a break at $\sim 50$ keV. For this reason, we use a broken power law model to better fit these data; the results are reported in Table 4. We build the SED in a broad energy range and plotted it in Figure 7. From low to high energy, data are from the 2Mass survey [62], WISE [63], SWIFT/BAT catalogs [64,65], EGRET catalog [66], AGILE catalog [67], Third Fermi catalog [68], and HAWC [46]. The SWIFT/XRT spectrum is extracted from an observation started on 2011-09-30T20:37:08 UTC, in photon count mode, in a 20 pixel radius circle. Table 4 reports the XRT/SWIFT fit results.

The proposed SED model does not favour any particular scenario. A SED constructed using simultaneously obtained data from radio to VHE, including INTEGRAL and CTA data, will be crucial in discriminating the physical process generating the non thermal photons at high energies. Figure 7 shows the sensitivity for the southern and northern sites of the CTA observatory (from https://www.cta-observatory.org/science/cta-performance/, accessed on 3 May 2021). CTA's unprecedented sensitivity between 20 GeV to 300 TeV will allow LS 5039 to be studied in more details at VHE and, in general, will allow us to delve into these sources like never before with detailed studies of the jet-medium interaction.

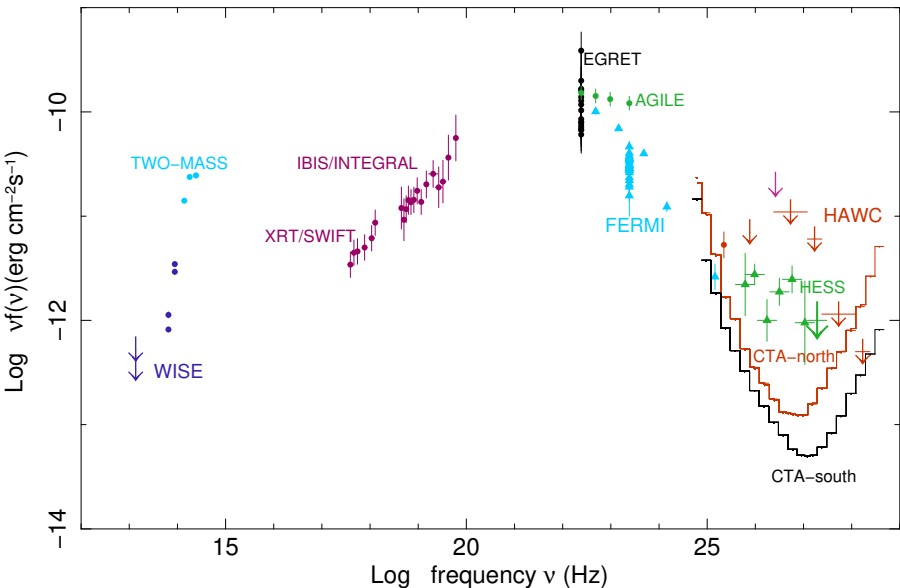

**Figure 7.** The spectral energy distribution of LS5039 using non-simultaneous data and showing the sensitivity expected from the southern (black line) and northern (orange line) CTA observatories **for 50 h** (black lines, from https://www.cta-observatory.org/science/cta-performance/, accessed on 3 May 2021). From low to high energy, the data used are from the: 2Mass survey (blue light points) [62], WISE (purple points) [63], EGRET catalog (black points) [66], AGILE catalog (green points) [67], Third Fermi catalog (blue light points) [68], HAWC (orange points) [46], and HESS (green points) [69].

### 3.2.2. PSR B1259−63

PSR B1259−63 is a $\gamma$-ray binary system composed of a rapidly rotating pulsar with a spin period of 48 ms and a bright O9.5V$_e$ stellar-type companion of $M_* \approx 30 M_{sun}$, at a distance of 1.5 kpc [70]. The binary system has a period $P_{orb} \sim 3.4$ years in an eccentric orbit ($e = 0.87$) and an orbital inclination angle $i_{orb} \approx 25$ degrees [71–73]. This source displays broad-band emission, which extends from radio wavelengths up to VHE $\gamma$-rays. In the radio domain, PSR B1259−63 shows a pulsed component detected near the periastron [71] and a transient unpulsed component far from periastron [74]. In the $\gamma$-ray band the source was periodically detected by the Fermi-LAT [75–77] and by the H.E.S.S. telescopes at the periastron passage [78–81]. H.E.S.S. observations conducted from 2014 to 2017 and Fermi-LAT observations from 2010 to 2017 have been analysed by the HESS Collaboration et al. [82]. These authors reported a periodic flux variability without a

clear detection of super-orbital modulation that was most probably caused by the changing environmental conditions near the periastron passage. Multiple flares of PSR B1259−63 have been detected before and after periastron by the Fermi-LAT instrument. The spectra obtained in both energy bands display a similar slope, although a common physical origin is ruled out.

The INTEGRAL satellite observed this source at the 2004 periastron passage. The IBIS/ISGRI spectrum is well fitted with a power law spectrum of photon index of $\Gamma = 1.3 \pm 0.5$ and an average luminosity of $(8.1 \pm 1.6) \times > 10^{33}$ erg s$^{-1}$ in the 20–80 keV energy band [83]. Several physical models have been proposed to understand the multi-wavelength emission. The favoured model is one considering the electrons that are accelerated by the shock between the pulsar and stellar wind [84–86]. Yi and Cheng [87] considered a complex model, according to which the GeV emission is due to Inverse Compton scattering of soft photons by the pulsar wind. The soft photons are from an accretion disk before the periastron while the density of the disk is not high enough after the periastron and the accretion is prevented by the pulsar wind shock. However none of the proposed models explain all of the features of the GeV emission.

### 3.2.3. Eta Carinae

The colliding wind binary system Eta Carinae includes two very massive stars and it is characterised from time to time by very energetic eruptions. The collision of strong stellar winds generates shocks that are expected to also produce cosmic rays. Non-thermal emission at X and $\gamma$-rays is produced by accelerated particles colliding with photons or with ambient matter. In recent years, the system has been observed with INTEGRAL [88,89], Suzaku [90], XMM-Newton and NuSTAR [90]. At $\gamma$-ray energies, detections have been reported with AGILE [91] and Fermi [92], while the H.E.S.S. telescope detected emission up to 400 GeV [93,94]. Although the analysis indicated a point-like morphology, the picture was not sufficiently clear and later observations did not indicate variability associated with lower energy measurements [94] and no flare similar to the one reported with AGILE has been revealed. The most recent oNuSTARbservation identifies Eta Carinae as the source of the high energy emission that is supported by flux variability with the binary orbital period reported before [95]. Additionally, the high energy emission connects to the soft-GeV and $\gamma$-ray spectrum with a power law slope of $\Gamma \sim 1.65$. This high energy emission seems not fully in agreement with the ones that were reported with Suzaku and INTEGRAL for which a larger flux was detected (see also [96]). Simultaneous measurements at different orbital phases are needed to further constrain the nature of the source emission.

### 3.2.4. LS I +61 303

The $\gamma$-ray binary LS I +61 303, located at a distance of $\sim$2 kpc [97], consists of a compact object and a B0-Ve star in an eccentric orbit ($e \simeq 0.7$) [98] of orbital period $P_{orb} \sim 26.5$ days [99] confirmed by INTEGRAL/IBIS long monitoring performed since 2002 up to 2008 in hard X-rays [43]. This source also exhibits a periodic superorbital modulation with a period of $\sim$4.5 years from radio [100] to X-ray [101] and GeV [102] bands.

The system, which wsa reported at VHE for the first time by MAGIC [103], is generally bright at TeV energies around apastron passage with flux levels between 1% and 25% of the Crab flux above 100 GeV [104–108]. VERITAS observed this source around apastron in 2014 when bright TeV flares were also seen and flux levels peaked above 30% of the Crab Nebula in less than one day [109]. The short timescale and the properties of the flares, in conjunction with the observed emission at 10 TeV during the flare, favour a micro-quasar scenario, although a jet that is produced in a pulsar binary cannot be ruled out. For these exceptionally bright TeV flares, Paredes-Fortuny et al. [110] present a pulsar wind shock scenario with an in-homogeneous stellar wind in which the star disc is disrupted, thereby increasing acceleration efficiency on a short timescale.

A young pulsar was at first suggested to be responsible for the observed radio emission [45], but no pulsations or spin period were ever detected, while the presence of long

quasi-periodic oscillations in radio and X-rays [111] supports a microquasar scenario. The observation of the correlated X-ray and TeV emissions and the non-correlated GeV-TeV emission imply that there are two distinct populations of accelerated particles producing the GeV and TeV photons [112]. Using non simultaneous radio, X-ray, and $\gamma$-ray observations, Massi et al. [113] study the emission along one single orbit. They report on a two peak profile in line with the accretion theory predicting two accretion-ejection events along the orbit. They also show that the positions of radio and $\gamma$-ray peaks are coincident with X-ray dips, as expected for radio and $\gamma$-rays, again supporting an ejection-accretion model. Similar to other $\gamma$-ray binaries, the nature of the compact object is unconfirmed and the physical SED models are not well understood. Because LS I +61 303 is a highly variable source, future simultaneous observations, including INTEGRAL and CTA, will be essential for the study of the physical processes at VHE.

### 3.3. Soft $\gamma$-ray to TeV Radiation in INTEGRAL AGN

In relation to the extragalactic sky, we have found 17 AGN, which are detected both at TeV and soft $\gamma$-ray energies by INTEGRAL/IBIS: 11 are BL Lac objects (nine high and two intermediate frequency peaked objects or HBL and IBL, respectively, as described in Table 1)[2], three Flat Spectrum Radio Quasars or FSRQ (PKS 1510−089, 4C +21.35 and 3C 279), and two FR I radio galaxies (Cen A and NGC 1275). One extra source, IGR J20569+4940 was up to now generically classified as a blazar, but of unknown class and redshift [114]. This source has only recently been announced as a TeV emitter [115] after observations with VERITAS, and it is also associated with a Fermi source (2FGL J2056.7+4939, [116,117]); it has a bright radio counterpart (NVSS 205642+494005) with a flat 2.7-10 GHz spectrum [118], as typically observed in radio-loud AGNs. It appears to be variable both in radio [119] and in X-rays [120]. Recently, using a Keck/LRIS spectrum, Clavel et al. [121] classified this source as a highly absorbed BL Lac object on the basis of a red and featureless continuum; the optical extinction is consistent with the high column density found in X-rays, which can be ascribed to both Galactic and intrinsic absorption.

In Figure 8, we assemble high energy data on this source using a NuSTAR observation that was performed in November 2015 with the average INTEGRAL and Fermi/LAT spectra with the addition of the reported TeV data: as can be seen in the figure the source SED probably has a peak in the GeV region. This SED resembles that of IGR J19443+2117 = HESS J1943+213 (see also Table 5), also in our list of associations, and one of the most extreme high peaked blazars shining through the Galactic plane. Therefore, we conclude that IGR J20569+4940 is another example of a high synchrotron peaked BL Lac, as originally proposed by Fan et al. [122]. This brings the number of HBL in our list to 10, which makes this class the most significant among TeV/INTEGRAL extra-galactic associations.

As recently suggested by Foffano et al. [123], the class of HBL may not be homogeneous, but made of two main sub-classes: one containing classical objects with very high energy $\gamma$-ray spectra peaking just below 1.0 TeV, and the other comprising hard-TeV sources peaking instead above 10 TeV. The former are probably those that are characterised in their SED by a high synchrotron peak above $10^{17}$ Hz, but with an inverse Compton hump peaking in the near TeV $\gamma$-ray band. These sources show moderate to high flux variability and, in some cases, display flaring activity. Conversely, the latter show a less variable flux, but an inverse Compton peak energy exceeding the 10 TeV threshold and a synchrotron peak near or exceeding 100 keV. In Table 5, we have collected SED data as well as IBIS/TeV spectral indices (errors include statistical and systematic uncertainties) for our small sample of high energy peaked BL Lacs to check their TeV sub-classification.

---

[2]   The two IBL are BL Lacertae and S5 0716+714

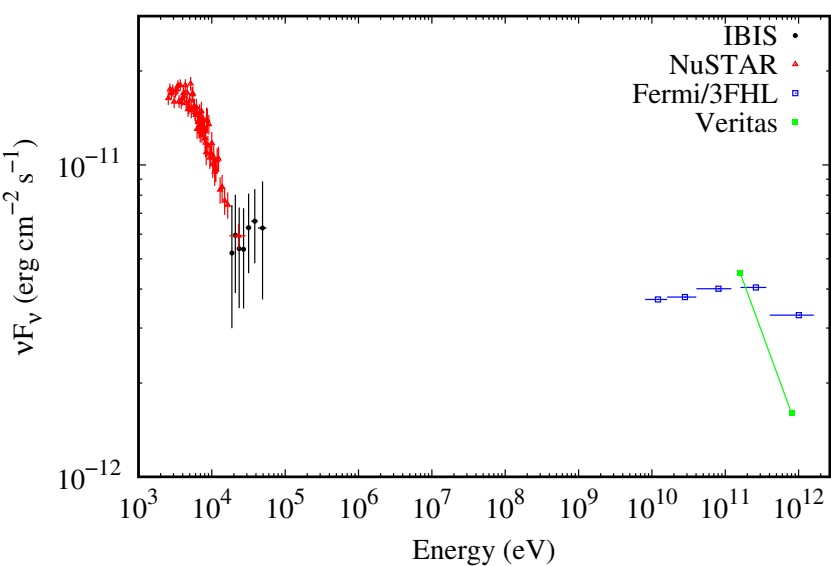

**Figure 8.** Spectral Energy Distribution of the blazar candidate IGR J20569+4940 **(see text)**.

**Table 5.** INTEGRAL/IBIS HBL.

| Source | Log Syn-Peak (ref) (Hz) | Soft $\gamma$-ray Slope S ‡ | Log Comp-Peak (ref) (Hz) | TeV Slope (ref) S ‡ |
|---|---|---|---|---|
| HESS J1943+213 | 18.1 (1) | $-0.58 \pm 0.68$(3) | $\sim$25.86 (4) | $-0.85 \pm 0.3$ (4) |
| MKN 501 | $17.7 \pm 0.2$ (2) | $-0.19 \pm 0.14$(3) | $25.68 \pm 0.1$ (2) | $-0.52 \pm 0.11$ (2) |
| 1ES 1959+650 | $17.4 \pm 0.2$ (2) | $-0.82 \pm 0.94$(3) | $25.68 \pm 0.2$ (2) | $-0.31 \pm 0.06$ (2) |
| 1H 1426+428 | $18.0 \pm 0.2$ (2) | $+0.23.77 \pm 0.77$(3) | $>27.38$ (2) | $+0.58 \pm 0.15$ (2) |
| MKN 421 | $17.0 \pm 0.3$ (2) | $-0.72 \pm 0.05$(3) | $26.23 \pm 0.1$ (2) | $-0.35 \pm 0.01$ (2) |
| 1ES 1218+304 | $17.9 \pm 0.2$ (2) | $-0.32 \pm 0.6$(3) | $25.68 \pm 0.3$ (2) | $-0.39 \pm 0.22$ (2) |
| 1ES 0033+595 | $17.9 \pm 0.2$ (2) | $-0.65 \pm 0.27$(3) | $\sim$25.38 (5) | $-1.8 \pm 0.70$ (5) |
| IGR J2056+496 | $17.0 \pm 0.2$ (2) | $-0.11 \pm 0.51$(3) | 25.98 (3) | $-0.77 \pm 0.4$ (8) |
| RX 1136.5+6737 | $18.2 \pm 0.2$ (2) | $-0.5 \pm 0.52$ (3) | 25.86 (6) | $-0.74 \pm 0.3$ (6) |
| TXS 0210+515 | $17.6 \pm 0.2$ (2) | $-0.76 \pm 0.66$ (3) | 25.68 (7) | $+0.0 \pm 0.30$ (7) |

‡ S: soft $\gamma$-ray and TeV slopes are related to the power-law spectral index S in the same energy band by S = 2.0 − Γ; **References**: (1) Chang et al. [124]; (2) Foffano et al. [123]; (3) this work; (4) Archer et al. [125]; (5) Aleksić et al. [126] (6) Hayashida et al. [127]; (7) Acciari et al. [128]; (8) Benbow and VERITAS Collaboration [115].

Where the extra objects discussed in Foffano et al. [123] have been plotted for comparison, all of our sources belong to the subclass of typical HBL, with the only exception of 1H 1426+428, as is evident in Table 5 and Figure 9. This source is unique, as its synchrotron peak is located above 100 keV, even if the source is in a low flux state [129], making it a very rare case of a hard-TeV BL Lac, as only four more such cases are known to date. It is worth noting (see Figure 9, right panel) that in the Compton peak versus soft $\gamma$-ray slope diagram the discrimination between typical and hard-TeV objects is more pronounced, which makes this diagram a more useful one to highlight the most extreme high energy peaked BL Lac objects.

Only three FSRQ are detected by the IBIS and TeV telescopes. In these cases, INTEGRAL spectra most likely probe the ascending side of the inverse Compton part of the SED contributing to establish its peak, while the very high energy $\gamma$-ray emission defines the final descending part of the same component.

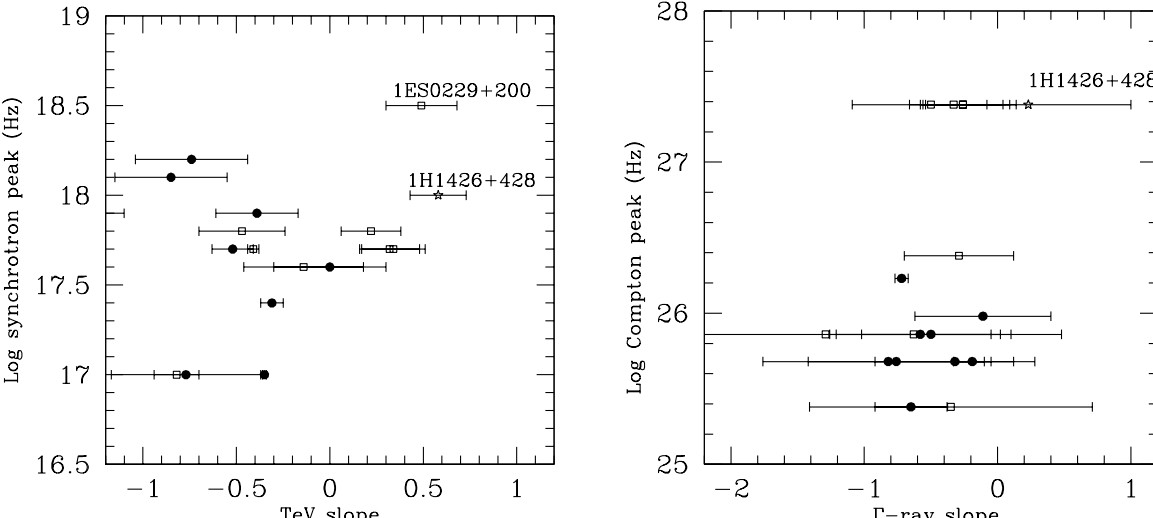

**Figure 9.** (**Left**): Synchrotron peak versus TeV slope; (**Right**): Compton peak versus soft $\gamma$-ray slope. Filled circles and star are sources reported in Table 5, i.e., HBL objects that are detected by INTEGRAL. Eight extra sources discussed in Foffano et al. [123], open squares, have been reported for comparison and their slopes have been evaluated using BAT 158-months spectra.

Typically, very high-energy (>100 GeV) emission is observed from this type of blazar either during flares or during their high activity states, although persistent emission during the low state was also detected. Historical TeV activity states can be found in MAGIC Collaboration et al. [130] and references therein for PKS 1510−089, in Albert et al. [131] and Emery et al. [132] for 3C 279, in Aleksić et al. [133] and Holder [134] for 4C 21.35. INTEGRAL IBIS, by covering the soft $\gamma$-ray band of the SED, therefore provides useful information for the understanding of emission models and offers the possibility of comparing variability studies in these two wavebands where the emission may have the same origin.

The TeV emission in all blazar type objects is believed to originate over rather small spatial ($\leq 1$ pc) scales, in the fast parts of the jet viewed almost pole-on where strong relativistic Doppler amplification favours their high energy detection. However in the radio galaxy Cen A, extended TeV emission has recently been reported [135], thanks to the unique proximity of the source. The estimated projected distance from the core is roughly a few kpc, although the real distance may be larger due to projection effects [136]. This observational result provides further evidence for the acceleration of ultra-relativistic electrons in radio jets and confirms their major role in the TeV emission and misaligned AGN. INTEGRAL/IBIS is not able to resolve different components within the same source, even in the case of Cen A, which is favourably located close by: the instrument angular resolution of 12 arcmin is not sufficient for observing extended emission at a level of a few arcminutes. However, indirect evidence for jet emission could come from broad band spectral analysis if the data are over a sufficiently wide energy range and of good statistical quality. The best analysis of the Cen A soft $\gamma$-ray spectrum using all INTEGRAL instruments was performed by Beckmann et al. [137]: they gave evidence for the presence of a curvature in the spectrum with an exponential cut-off at around 400 keV, which is likely associated with thermal Comptonisation in the hot corona, around the nucleus. Extending this Comptonisation model to the GeV range shows that it cannot explain the high-energy emission, implying the existence of an extra component of non-thermal origin probably related to the jet. At the moment, the relative weight of these two components (thermal versus non thermal) within the soft $\gamma$-ray waveband is still unknown, but further INTEGRAL observations of the source combined with variability studies of its soft $\gamma$-ray emission can help in solving this issue.

### 3.4. INTEGRAL Counterparts of Unidentified TeV Sources

The online TeV catalogue lists ∼230 TeV sources (as of January 2021) and the nature of the majority of them has been firmly established through multi-wavelength observations. Interestingly the fraction of unidentified TeV sources, with no firmly established counterparts at other wavebands, is about 25%. They lie essentially on the Galactic plane (only four unidentified sources, out of a total of 54, are far from the plane), although this could be due to an observational bias. In fact, to date, a large portion of the entire Galactic plane has been covered by observations of the current IACTs, although with non-uniform exposure. In particular, the best surveyed regions, in terms of exposure and sensitivity, are those in the inner Galactic plane at longitudes approximately from l = 250° to l = 65°, thanks to the Galactic plane surveys performed by the IACT array H.E.S.S. [138]. In the light of the above, it can be inferred that the great majority of unidentified TeV sources are galactic objects within the Milky Way. As examples, to date, the most common TeV sources firmly identified with Galactic objects mainly include pulsar wind nebulae, supernova remnants, and high mass X-ray binaries (HMXB).

The identification of unknown TeV sources is crucial for obtaining a deep insight into the nature of the source population at TeV energies. In addition, it is most probably among unidentified objects that peculiar sources, or even a new class of sources, could emerge. However, one of the main difficulties in the identification process is the often large error box of TeV sources, hence positional correlation with known objects is not usually enough to firmly identify the TeV nature of the source. For this reason, a multi-wavelength approach is strongly needed to understand their nature. Searches for X-ray counterparts, especially above 20 keV, are particularly useful in finding a positionally-correlated, best candidate counterpart with parameters that might be expected to produce $\gamma$-rays at GeV-TeV energies. Furthermore, information from the soft $\gamma$-ray band is very useful in characterising these sources in terms of spectral shape, flux, absorption properties, and variability. In the following sub-sections, we highlight our findings from positional correlations that are found between some unidentified TeV sources and soft $\gamma$-ray objects detected by INTEGRAL above 20 keV.

#### 3.4.1. HESS J1841−055

HESS J1841−055 is an unidentified source discovered at TeV energies by H.E.S.S. in 2007 during the Galactic plane survey [139]. Remarkably, its emission was observed as highly extended. This characteristic has been confirmed by the more recent H.E.S.S. results published in 2018 from the latest Galactic plane survey [138]. The source has been studied above 1 TeV by other experiments such as ARGO-YBJ [140] and HAWC [141]. From the Fermi-LAT catalogue of Galactic extended sources [142], in this region there are two extended sources above 10 GeV, whose emission is overlapping with the extension at TeV enegies. Recently, the MAGIC collaboration [143] reported a deep study of HESS J1841−055 using MAGIC and Fermi-LAT data. They fully confirmed the diffuse and significantly extended emission from the source, with an estimated size (∼0.4 degrees) that is similar to that previously measured by H.E.S.S. In addition, there are several bright and highly significant hot spots in the diffuse TeV emission that strongly hint at the presence of multiple TeV sources in the region. This indicates that probably several point-like sources are contributing to the extended high energy emission from HESS J1841−055. In particular, MAGIC Collaboration et al. [143] concluded that such emission can be best interpreted by scenarios, like PWN and SNR.

The source sky region has been further investigated at much lower energies to search for the possible best candidate counterparts. Although no firm counterparts have been identified, several likely associations have been proposed in the literature. To date, the most promising associations are those involving the pulsar PSR J1838−0537 and the SNR G26.6−0.1, based on spatial and energetic considerations. In addition, the point-like source AX J1841.0−0536 is an interesting candidate counterpart,, as reported by Sguera et al. [144], making use of INTEGRAL data in both energy bands 3–10 keV and 20–100 keV. In fact,

AX J1841.0−0536 is the only soft $\gamma$-ray source detected by INTEGRAL above 20 keV inside the error region of HESS J1841−055. As suggested by Sguera et al. [144], AX J1841.0−0536 could be responsible for at least a fraction of the entire TeV emission from the extended source HESS J1841−055. This proposed association, based on a striking spatial match, is also supported from an energetic standpoint by a theoretical scenario, where AX J1841.0−0536 is a low magnetised pulsar which, due to accretion from the super-giant companion donor star, undergoes sporadic changes to transient Atoll-states (e.g., [145]) where a magnetic tower can produce transient jets and, as a consequence, high-energy emission. The proposed association is interesting, because AX J1841.0−0536 might eventually be the prototype of a new class of Galactic TeV emitters. In fact, it is a source firmly identified as a member of the newly discovered class of transient HMXBs, named as Supergiant Fast X-ray Transient (SFXTs, [146,147]). It is noteworthy that the possible association between the SFXT AX J1841.0−0536 and HESS J1841−055 might not be a unique and rare case. In addition, other interesting associations between SFXTs and unidentified high energy sources have been proposed in the literature [148,149].

Instruments with much better angular resolution than the current generation of very high energy telescopes is required to disentangle all of the proposed counterparts associated with HESS J1841−055 at TeV energies. This source is naturally an interesting target for further studies with the forthcoming CTA due to its excellent angular resolution (expected to be better than two arcminutes at energies above several TeV) as well as source localisation accuracy (namely, aimed for 10 arcseconds).

### 3.4.2. HESS J1844−030

Most of the TeV sources that are detected by H.E.S.S. during its survey of the Galactic plane are extended. Point-like TeV sources, i.e., having a spatial extension on the scale of the few arcminutes resolution of H.E.S.S., are the exception. HESS J1844−030 is an unidentified point-like TeV source, as reported in the last release of the H.E.S.S. Galactic plane survey [138]. It is spatially associated with the radio source G29.37+0.1 and the candidate X-ray pulsar wind nebula (PWN) G29.4+0.1. Castelletti et al. [150] performed a multi-wavelength study of the source region, showing that G29.37+0.1 is a radio source with a complex morphology that could be due to the superposition of two different and unrelated objects: a background extragalactic source (likely a radio galaxy) and a foreground galactic source (likely an SNR). Because of the particularly complex morphology of G29.37+0.1, the authors were not able to disentangle the origin of the high energy nature of HESS J1844−030. Recently Petriella [151], using radio and X-ray data, presented strong evidence that G29.4+0.1 is a PWN that is powered by a point-like X-ray source embedded in it. The author proposed that HESS J1844−030 is the high energy counterpart of G29.4+0.1, supporting this association with a leptonic mechanism to explain the observed TeV emission.

The sky region of HESS J1844−030 has been covered by INTEGRAL observations with an exposure of 5 Ms, according to the latest published catalogue of Bird et al. [9]. Figure 10 shows the corresponding IBIS/ISGRI deep significance mosaic map of the source sky region (20–40 keV). As can be seen, only one persistent soft $\gamma$-ray source has been significantly detected by INTEGRAL ($7\sigma$ level, 20–40 keV), which is spatially associated with the positional uncertainty region of HESS J1844−030. As such, it is the best candidate counterpart in the soft $\gamma$-ray band. The measured 20–40 keV flux is 0.6 mCrab or $4.5 \times 10^{-12}$ erg cm$^{-2}$ s$^{-1}$. The source detected by INTEGRAL has been named AX J1844.7−0305 [9], since the latter is the closest catalogued X-ray object. A further in-depth investigation of the INTEGRAL data on AX J1844.7−0305 is currently under way, to definitely confirm the proposed physical association with HESS J1844−030. The collected information in the soft $\gamma$-ray band could be useful for characterising HESS J1844−030 in terms of variability, absorption properties, and spectral shape, hence to shed more light on its nature.

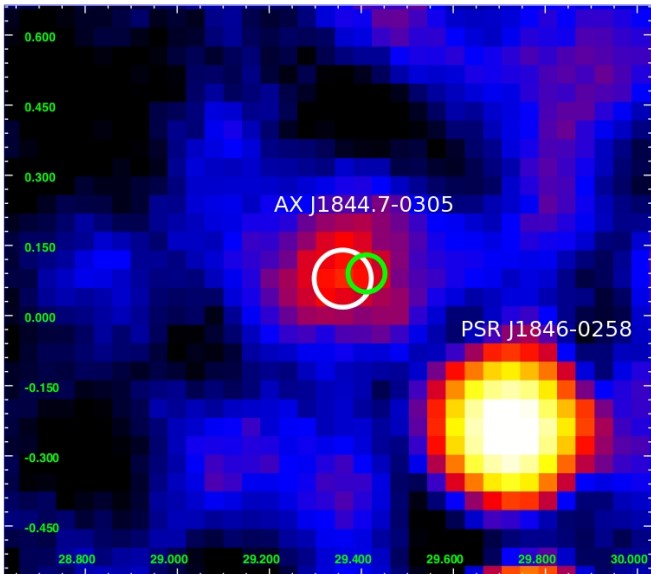

**Figure 10.** INTEGRAL/IBIS mosaic significance image (20–40 keV, 5 Ms exposure time, Galactic coordinates) of HESS J1844−030 sky region. The white circle (90% confidence radius of 3.7 arcminutes) represents the positional uncertainty region of the source AX J1844.7−0305 detected by INTEGRAL ($7\sigma$ level). The green circle (95% confidence radius of 2.4 arcminutes) represents the positional uncertainty region of the point-like TeV source HESS J1844−030. The other very bright source that is detected by INTEGRAL in the image is the PWN around PSR J1846−0258.

### 3.4.3. HESS J1808−204

HESS J1808−20 is an unidentified TeV source that is characterised by steady and extended emission. Based on a striking spatial correlation, potential counterparts to the high energy emission are the massive stellar cluster Cl* 1806−20, the luminous hypergiant star LBV 1806−20, and the soft $\gamma$-ray repeater SGR 1806−20 [138]. Although not firmly established yet, all such associations are very interesting, because they provide intriguing potential for high energy particle acceleration from peculiar sources. In fact, Cl* 1806−20 is a massive stellar cluster hosting numerous energetic stars, such as Wolf-Rayet stars, OB super-giants, and, in particular, the rare luminous blue variable hyper-giant LBV 1806−20 that generates a very powerful wind and it is among the most massive and luminous stars known in our Galaxy. Interestingly, SGR 1806−20 is also a member of the cluster Cl* 1806−20. It is a strongly magnetised neutron star belonging to the class of magnetars whose soft/hard X-ray emission is powered by the decay of their huge magnetic field. In particular, SGR 1806−20 is known to be an active source of short and energetic soft-$\gamma$-ray flares, especially famous for its 2004 December very giant flare with an energy release of several $10^{46}$ erg s$^{-1}$, i.e., one of the strongest outbursts ever recorded at soft $\gamma$-rays from any known soft gamma-ray repeater [152]. From an energetic stand-point, in principle, the TeV energy flux of HESS J1808−204 could be produced by the energetic members of the massive star cluster Cl* 1806−20 through particle acceleration that results from the stellar wind interaction over parsec scales according to the cluster size and stellar density [138,153]. In particular, the member LBV 1806−20 could dominate much of the stellar wind energy and, therefore, drive most of the particle acceleration. An intriguing possibility is that SGR 1806−20 could contribute to an additional component of emission from HESS J1808−204 through inverse Compton scattering from a PWN powered by the magnetic field rapid decay, similarly to the case of HESS J1713−381 and the magnetar CXOU J171405.7−-381031 [154]. A major uncertainty of such a leptonic scenario is that X-ray observations have, so far, failed to identify a PWN in the region.

We note that SGR 1806-20 is the only persistent soft $\gamma$-ray source that is detected by INTEGRAL inside the error region of HESS J1808-204 in the broad energy range 20–100 keV, based on a total of about 8 Ms of exposure time observations according to the latest

INTEGRAL/IBIS catalogue [9]. The measured fluxes are equal to $2 \times 10^{-11}$ erg cm$^{-2}$ s$^{-1}$ (20–40 keV) and $3.2 \times 10^{-11}$ erg cm$^{-2}$ s$^{-1}$ (40–100 keV). Clearly, the source is characterised by a soft $\gamma$-ray tail significantly extending up to at least 150 keV [155]. The emission has a power law spectrum with a photon index in the range 1.5–1.9 and a 20–100 keV flux of $5 \times 10^{-11}$ erg cm$^{-2}$ s$^{-1}$ (see the spectrum in Mereghetti et al. [155]), corresponding to a luminosity of $1.3 \times 10^{36}$ erg s$^{-1}$ at 15 kpc distance. INTEGRAL observations of magnetars provided the first ever detection of persistent emission in the energy range 20–200 keV and opened a new important diagnostic to study the physics of magnetars.

## 4. Summary

We have shown that, as expected, there is a significant correlation between the recent INTEGRAL/IBIS 1000 orbit catalogue and the online TeV source list. By analysing this correlation, we have further shown that 39 objects ($\sim$20% of the VHE $\gamma$-ray catalogue) have emission in both the soft $\gamma$-ray and TeV wavebands, providing an indication of the usefulness of combining information at these frequencies. The objects discussed belong to various classes, both galactic and extra-galactic, as well as unclassified sources. In the galactic realm, compact objects, like binary systems, pulsars, and extended objects, like SNR and PWN, are reported and discussed in detail. In the extra-galactic case, AGN of various classes have been found and investigated, like the two types of Blazars (BL Lac and FSRQ), as well as radio galaxies. Finally, the identification of objects that are still lacking a definite counterpart at TeV energies can benefit from information at soft $\gamma$-ray frequencies. Overall, by discussing the individual object class, we have further emphasised the importance of adding to the broad knowledge of TeV sources information in the soft $\gamma$-ray waveband. With this in mind, the INTEGRAL legacy (19 years of measurement collection with spectral, timing, and imaging information) will provide one of the most extended databases to be exploited, particularly on the galactic plane, where many TeV sources are located. These data are made available to the astrophysical community by means of various channels: through the on line archive service at the ISDC (https://www.isdc.unige.ch/integral/heavens, accessed on 3 May 2021), by means of specific on going projects, like, for example, the galactic plane survey that makes data publicly available almost in real time (http://gps.iaps.inaf.it/, accessed on 3 May 2021) and direct contact with the teams in charge of INTEGRAL surveys (any of the authors of this paper for INTEGRAL 1000 orbit catalogue products).

At the moment, INTEGRAL is operating in nominal mode with all of the instruments working well. The mission is approved for operations till the end of 2022, with a possible further extension until 2025. If approved, this will give an opportunity to perform high sensitivity simultaneous investigations in the keV to TeV energy range in the next half of the decade. As such, it provides an observational opportunity for currently operational VHE telescopes, while building, at the same time, a strong legacy for future projects, such as the CTA.

**Author Contributions:** Authors of the present paper are part of the IBIS survey team. As such they all contributed to build up images, mosaics, light curves and spectra of all sources included in the IBIS catalogues constructed over the years. They are also deeply involved in the search of counterparts of the unidentified detected sources from INTEGRAL/IBIS and SWIFT/BAT. They are all on the way to perform a new survey that will be the legacy for the future observation with the CTA project. Conceptualization, all authors have contributed after previous work done together on the same argument; methodology, all authors have discussed the analysis method after several discussions; software for cross correlation analysis and results, J.B.S.; formal analysis, investigation and writing of the original draft preparation, all authors with separate responsibility for individual sessions (Section 1: P.U., L.B.; A.J.B.; Section 2: J.B.S.; Section 3.1: L.B., A.M., L.N., E.P.; Section 3.2: A.B., M.F.; Section 3.3: A.M., E.P.; Section 3.4: V.S.; summary: A.J.B.); data curation, all authors have contributed within their specific expertise (see above); writing—review and editing, A.M., J.B.S., P.U.; visualization, A.M., J.B.S., M.F., V.S.; supervision, A.M.; funding acquisition, A.M., L.N. All authors have read and agreed to the published version of the manuscript.

**Funding:** The research by Italian co-authors was funded by different agreements between the Italian Space Agency (ASI) and INAF over the past 20 years, last of which is 2019–35-HH.0. AJB

acknowledges partial support for this research to the European Union Horizon 2020 Programme under the AHEAD project (grant agreement n. 654215).

**Data Availability Statement:** All data reported in the paper are publicly available on the catalogues linked throughout the text.

**Acknowledgments:** The authors would like to acknowledge the contribution made in the field of INTEGRAL/TeV associations, particularly PWN, by their friend and colleague A.J. Dean, who has played over the years an inspiring role for many scientists.

**Conflicts of Interest:** The authors declare no conflict of interest.

## Abbreviations

The following abbreviations are used in this manuscript:

| | |
|---|---|
| AGILE | Astro-Rivelatore Gamma a Immagini Leggero |
| AGN | Active Galactic Nuclei |
| Blazar | Blazing quasi-stellar object |
| BH | Black Hole |
| CTA | Cherencov Telescope Array |
| CR | Cosmic Ray |
| EGRET | The Energetic Gamma Ray Experiment Telescope |
| ESA | European Space Agency |
| FERMI | Fermi Gamma-ray Large Area Space Telescope |
| FR I | Fanaroff-Riley Class I |
| FSRQ | Flat Spectrum Radio Quasars |
| HAWC | High-Altitude Water Cherenkov Observatory |
| HESS | The High Energy Stereoscopic System |
| HMXB | High Mass X-ray Binary |
| HBL | High Peaked Bl Lac |
| kyr | kilo year |
| IBIS | IMAGER on Board the INTEGRAL Satellite |
| IC | Inverse-Compton |
| IBL | Intermediate peaked Bl Lac |
| INAF | National Institute for Astrophysics |
| INTEGRAL | INTErnational Gamma-Ray Astrophysics Laboratory |
| ISGRI | INTEGRAL Soft Gamma Ray Imager |
| LMXB | Low Mass X-ray Binary |
| mCrab | milliCrab |
| MAGIC | Major Atmospheric Gamma-ray Imaging Cherenkov Telescopes |
| MDPI | Multidisciplinary Digital Publishing Institute |
| Ms | Million seconds |
| NS | Neutron Star |
| NuStar | Nuclear Spectroscopic Telescope array |
| PSR | Pulsar |
| PWN | Pulsar Wind Nebula |
| QSO | Quasi Stellar Object |
| SED | Spectral Energy distribution |
| SGR | Soft gamma Ray Repeater |
| SN | Supernova |
| SNR | Supernova Remnant |
| SPI | Spectrometer on INTEGRAL |
| SMBH | Super Massive Back Hole |
| SFXT | Supergiant Fast X-ray Transient |
| SWIFT/BAT | Gamma Ray Burst Explorer/Burst Alert Telescope |
| TeV | Tera electron Volt |
| VHE | Very High Energy |
| XMM | X-ray Multi Mirror |
| WISE | Wide-field Infrared Survey Explorer |

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
