# Peer review of "INTEGRAL View of TeV Sources: A Legacy for the CTA Project"

_universe, doi:10.3390/universe7050135_

Round 1

Reviewer 1 Report

The present manuscript describes the INTEGRAL observation legacy on a fraction of the current observational dataset for the upcoming IACT, i.e., CTA. The paper is well written and provides an interesting overview of the relevant connection between INTEGRAL and TeV source detections in order to probe the underlying particle acceleration mechanisms in sources. It summarizes  the current knowledge on specific case cases. I have listed my comments and suggestions that should be addressed below before I can recommend it for publication.

General comments:

The input of the upcoming CTA measurements should be more detailed with more quantitive explanations on how CTA upcoming measurements will bring new insights to the current knowledge, in particular how these measurements would help to disentangle between the currently-viable particle acceleration models. Figure captions should be significantly expanded in order to clearly detail the content of the plot. In addition, some proposed models derived to explain the measurements should be over plotted in order to better highlight what would be the CTA capabilities. 

Other comments and suggestions:

Abstract: one should clarify that the present study is not the final INTEGRAL legacy to CTA since only a fraction of all currently-available data is considered here.

Introduction : mention the main characteristics like flux sensitivity, angular and energy resolution for the relevant instruments, i.e., INTEGRAL and IACTs, including expected CTA performances.

l11. Fermi-LAT is a satellite experiment, H.E.S.S. should be written with dots everywhere in the paper draft

l26: please quantify what is meant by « the deepest survey » 

Fig.1: 

- why the GC region is not shown ? 

- add legend on the axes and color scale, labels can be barely read

l31: define VHE once and use it later 

l50: more details on the cross-correlation analysis would be helpful

l60: mention the position uncertaintties of IBIS and TeV sources. Is the position uncertainty accounted in the cross-correlation analysis ?

l88: VERITAS should be mentioned too

Tab.1: please mention the (approximated) position uncertainty of the INTEGRAL sources

Fig. 3: caption: what are the solid line and points. Provide more details. 

l108-109: does the error bars include the statistical and systematic uncertainties ?

l111: what is the significance of this excess ?

l118-119: mention the possible energy cut-off detected in the Cas A TeV gamma-ray spectrum

Fig.4: Ref. to ROSAT and HESS papers needed

Tab2:  do the error bars on the spectral index include the statistical and systematic uncertainties ?

Fig. 7. Legend needs to be be significantly improved. HESS measurements should be plotted. Plotting current viable models according to measurements would help. Please mention how many hours are assumed to plot the CTA sensitivity.

l276-281: why H.E.S.S. measurements are not used?

l282: one would like to see the SED model and understand with more details how CTA measurements could help

Tab. 5: do the error bars on the spectral index include the statistical and systematic uncertainties ?

Fig. 9 :add error bars here

3.4: one should emphasize how much the Galactic plane is covered by the current IACTs

L482-483: one should quote the source position uncertainty expected for CTA which is the relevant quantity for source identification

Summary : mention when the 19-year INTEGRAL legacy study is expected

Reviewer 2 Report

Main comment is that it's not currently clear why the 1000 orbit catalogue is used which only includes data until 2010. Although not published as catalogues, it seems that much more data is available. 
As you state, this "therefore only gives a glimpse of the possible associations" --> so why not use a more extensive dataset? 

Other minor comments:

Figure 1: Axes, colour bar labels and source labels are not easily readable – please increase the font size

L45: note that not all of the 229 objects listed by TeVcat are unique sources – there are many cases, particularly along the Galactic plane, of an extended source as seen by one instrument (e.g. HAWC) listed separately from coincident emission seen by another instrument, simply because the view of one instrument overlaps with multiple other sources that cannot be distinguished. Examples:  HESS J1813—178 and 2HWC J1814-173; HESS J1857+026, MAGIC J1857.2+0263 and MAGIC J1857.6+029 etc. etc.

Please add a comment on whether or not this was taken into account in your study / how this would modify the results or similar.

L53-55, L68: shouldn’t the “correlation within a specified angular distance” also be adjusted based on the TeV source extension? (instead of a hard cut at 330 arcsec, e.g. a “1sigma” cut…)

Figure 2: coordinate labels missing (left) and plots are too small.

Would suggest the following changes:
Left plot: TeV and INTEGRAL on same plot but with different colours/marker shapes (easier to see extragalactic correlations by eye)

Right plot: use same axis. Could merge, such that lower right is a bottom panel of the same plot - as is usual for residuals. (It is after all just the difference of the above curves.)

Plots need to be larger.

L69-72: clarify if this increases the date range / number of orbits?

L74: remove “at”

(L75-76: again, seems odd – why not use all 19 years in this study? )

L88: add VERITAS

Table 1: not “versus”, rather something like: “Association of TeV sources with INTEGRAL/IBIS”

Figure 3: Plot size and font size again far too small.

The comparison would be more illuminating if the two spectra were plotted together in a single plot. (With different colours for Tycho / CasA as indicated in a legend)

L111: could these titanium lines be indicated in the figure 3?

L112: “do not have the sufficient” --> “do not have sufficient”

L120: is “44Ti emission” the same as “Titanium” above?  please clarify / Use consistent terminology

Figure 4: Please label somewhere on the figure that the maps are in RA-Dec coordinates. 

L174: missing “J” in “PSR J1420”

L186: missing “J” in “HESS J1616-508”

Figure 5: Difficult to compare the HESS centroids to the INTEGRAL map by eye. Is it possible to overlay the positions of the two HESS sources on the INTEGRAL map?

L215: missing “J” in PSR J1846”

L223: “that that” --> “than that”

Table 3: for pulsar properties, the reference should not be the HESS collaboration paper, but rather the ATNF catalogue.

L243: “can derived” --> “can be derived”

General comment: please be consistent with “cross-correlation” or “cross correlation”. Should presumably be the former throughout.

L262: see also Volkov et al, arxiv:2103.04403

L268: please format these references as [45,46,47,48] or [45-8], and not ([45],[46],[47],[48])

Figure 6: Please correct the source names in the caption. Add a dotted line at 0 in the residual plot. Y-axis labels of 10^-2 and 4 overlap – illegible. Perhaps the y-axis of the bottom panel could be labelled “residual”. Top panel – not all points are visible on the plot, e.g. LSI+61 303.
Models don’t look like particularly good fits…

Table 4: It might be worth adding here an indication of the goodness of fit E.g. chi^2/ndf

L286-7: this needs rephrasing; LS 5039 has already been detected at VHE. (reads as if CTA will be first) presumably you mean “studied in more detail”

L296 – 7, L308: formatting of references (see comment above)

Figure 7: please distinguish the two black curves (two different colours or plotting styles).

Add a legend or indicate all instruments – it is not clear which points are Fermi / EGRET etc.
This should include HESS data e.g. HESS collaboration, Science, 309, 746-749 (2005)

Stray apostrophe after caption.

L312: explains --> explain

L320: HESS has 5 telescopes – would be clear here e.g. “the largest telescope of the H.E.S.S. array”

L331: missing space in binary name.

L338: word order --> “during which bright TeV flares were seen

L353: “is in doubt” --> “is unconfirmed”

L354: understand --> understood; high variable --> highly variable

Figure 8: plot is a bit small. Does it make sense to plot in terms of frequency, when it is later discussed in terms of energy?

Table 5: Please turn the caption into a sentence.

L413: delete “as said”

Figure 9: plots can be larger, remove white space between plots and above caption.
typo: peack --> peak in y-axis title, and “sync” --> Sync. or synchrotron

L444: “in the following…” in the following what? Presumably, “in the following sub-sections” or “in the remainder of this section”…

L456: estimated value --> estimated size.

L474: the term “Atoll-states” isn’t widely known – this is described in the rest of the sentence? Maybe cite a definition.

Figure 10: add source names to the figure.
caption: “PWN PSR” doesn’t make a lot of sense. Would rather say “is the PWN around PSR J1846-0258, associated with the TeV source HESS J1846-029.

Section 3.4.3 (last paragraph) – would be interesting to also see a spectrum / SED of this source.

L584: stray dot
